# Preparation of Polyimide/Ionic Liquid Hybrid Membrane for CO_2_/CH_4_ Separation

**DOI:** 10.3390/polym16030393

**Published:** 2024-01-31

**Authors:** Xiaoyu Du, Shijun Zhao, Yanqing Qu, Hongge Jia, Shuangping Xu, Mingyu Zhang, Guoliang Geng

**Affiliations:** College of Materials Science and Engineering, Qiqihar University, Qiqihar 161006, Chinavipquyanqing@163.com (Y.Q.);

**Keywords:** polyimide, gas separation membrane, ionic liquid, separation permeability

## Abstract

Imidazole ionic liquids (IL_s_) have good affinity and good solubility for carbon dioxide (CO_2_). Such ionic liquids, combined with polyimide membrane materials, can solve the problem that, today, CO_2_ is difficult to separate and recover. In this study, the ionic liquid (IL) of 1-ethyl-3-methylimidazolium tetrafluoroborate (IL_1_), 1-pentyl-3-methylimidazolium tetrafluoroborate (IL_2_), 1-octyl-3-methylimidazolium tetrafluoroborate (IL_3_), and 1-dodecylimidazolium tetrafluoroborate (IL_4_) with different contents were added to a polyimide matrix, and a series of polyimide membranes blended with ionic liquid were prepared using a high-speed mixer. The mechanical properties and gas separation permeability of the membranes were investigated. Among them, the selectivity of the PI/IL_3_ membrane for CO_2_/CH_4_ was 180.55, which was 2.5 times higher than the PI membrane, and its CO_2_ permeability was 16.25 Barrer, which exceeded the Robeson curve in 2008; the separation performance of the membrane was the best in this work.

## 1. Introduction

The increase in carbon dioxide (CO_2_) in the atmosphere is mainly due to the release of flue gas, the burning of natural gas, etc. [1,2,3,4]. Meanwhile, the existence of CO_2_ reduces the exploitation value of natural gas, affects the combustion efficiency of methane, increases cost, and leads to pipeline corrosion [5,6]. Because of this, recovering carbon dioxide from industrial waste gases is crucial for both future use and environmental considerations [7]. Amine absorption is a cutting-edge technology for removing CO_2_, but it has high capital and operating costs, a sophisticated operating method, and will negatively impact the environment [8,9,10]. As opposed to conventional methods such as amine absorption, membrane separation technology offers the distinct advantages of being more environmentally friendly, requiring less upkeep, and having a smaller environmental impact [11,12]. The separation efficiency of amine absorption is low compared to the separation efficiency of membrane separation in the aforementioned methods for mixed gas [13,14]. 

Due to their excellent processability, low cost, and accessible availability, polymer-based membranes are currently widely applied in gas separation [15,16]. Due to their superior thermal, chemical, and mechanical qualities, as well as their exceptional film-forming abilities, polyimides have shown strong attraction embodied in gas separation as membrane materials. Numerous studies [17,18] have been conducted to determine that the chemical makeup of polyimides affects their gas penetration characteristics. However, the PI membrane is influenced by the “trade-off” effect between permeability and selectivity [19,20,21], and it is challenging to overcome the Robeson upper bounds [22,23]. As a consequence, there are numerous strategies by which to increase gas permeability and selectivity, including molecular design [24,25], inorganic inclusion [26,27], chemical crosslinking [28], and the blending of organic species [29,30,31,32,33], among others. In molecular design, the synthesis of structures with specific impacts on the gas molecular structure, such as the spiral ring structure and bridge ring structure, would increase the gas permeability of the membrane. The permeability of the membranes is enhanced by adding a variety of modified particles, which enhance the free volume between polymers. As a result, this article focuses on how to enhance the performance of the membrane in gas separation.

During the past 20 years, ionic liquids (ILs) have received widespread attention due to their new, tunable physicochemical properties [34]. These unique properties and functions have become a new type of compound used for the development of advanced multifunctional materials, with outstanding potential in several fields of application. In composite materials, the combination of ILs and polymers has helped to develop different types of intelligent materials, which synergistically combine the characteristics of specific polymers and ILs [35]. In addition, ILs can be widely modified by incorporating functional groups with specific properties into cations, anions, or both. Therefore, ILs, polymers, or both can be adjusted to obtain a wide range of multifunctional composite materials and meet the specific requirements of many applications. In recent years, ionic liquids have also been widely used in the field of gas permeation, especially ionic liquids containing imidazole groups. Through research, it has been found that imidazole-based ILs have an adsorption effect on CO_2_, which can improve CO_2_ solubility, diffusion rate, and permeability, and are used for gas separation applications. The advantage of the beneficial qualities of the components, that blending additives into the base polymer material, is one of the most promising methods by which to increase the gas separation performance [11]. Due to their distinctive characteristics, such as low vapor pressure and good thermal stability, ionic liquids (ILs) have been regarded as promising absorbents for CO_2_ separation [34,35]. Ionic liquids (ILs) have recently been mentioned in several studies as membrane additives for separation [36,37,38,39,40]. Shinji Kanehashi created composite membranes by casting glassy fluorine-containing polyimide (PI) with up to 81 weight percent of 1-butyl-3-methylimidazolium bis(trifluoromethylsulfonyl)imide ([BMIM][Tf_2_N]) ionic liquid (IL) [40]. The PI + IL membranes with >51 wt% IL had greater gas permeability. The higher CO_2_ dissolution selectivity greatly enhanced the gas selectivity at 0 °C as well. These composite membranes offer greater potential to improve the CO_2_ separation performance than conventional, general PIs.

The addition of different types of concentrations of ionic liquids with the polyimide matrix is a useful technique; we describe how to enhance the separation performance of the membrane for mixed gas in the current study. The physical modification of polyimide film was carried out by incorporating different types and contents of ionic liquids into the polyimide matrix. The gas separation permeability and mechanical properties of the membrane were tested and analyzed. Therefore, in this article, four types of ILs with different alkyl chain lengths were introduced into polyimide (ODA-6FDA) substrates to prepare PI/ILn (x wt%) blend membranes with IL contents of 5%, 10%, 15%, and 20%. The mechanical properties and the gas permeability of the CO_2_ and CH_4_ of the membranes were mainly studied. The prepared thin membrane was tested using a tensile strength tester and a gas permeability tester. The effects of different types and concentrations of ionic liquids and the effects of different polyimide substrates on the membrane separation permeability were explored. The study discovered that the prepared polyimide membrane had better separation performance when the ionic liquid was 1-octyl-3-methylimidazolium tetrafluoroborate (IL_3_) and the ionic liquid content was 15%, opening up research opportunities for addressing significant issues pertaining to membrane separation technology.

## 2. Experimental Sections

### 2.1. Experimental Materials

The diamine used was 4,4′-diamino diphenyl ether (ODA) (purity > 98%), the anhydride used was 4,4′-(hexafluoroisopropylidene) diphthalic anhydride (6FDA) (purity > 99%). All of the aforementioned were purchased from Aladdin Biochemical Technology Co. Ltd. (Shanghai, China), while N’N-Dimethylformamide (DMF) was bought from Combo Chemical Reagent Co. Ltd. (Tianjin, China) (purity > 98%). The ionic liquids necessary for this study were 1-ethyl-3-methylimidazolium tetrafluoroborate (IL_1_) (purity > 98%), 1-pentyl-3-methylimidazolium tetrafluoroborate (IL_2_) (purity > 98%), 1-octyl-3-methylimidazolium tetrafluoroborate (IL_3_) (purity > 98%), and 1-dodecylimidazolium tetrafluoroborate (IL_4_) (purity > 98%), which were purchased from Qingdao Aolike New Material Technology Co. Ltd. (Qingdao, China).

### 2.2. Experimental Steps

ODA (0.40 g, 2.00 mmol), 6FDA (0.91 g, 2.04 mmol), and DMF (5.50 mL) were added to a three-necked flask and agitated for 4–5 h to ensure sufficient reaction. The synthesis of PI membranes is shown in Appendix A. Because 6FDA is easily hydrolyzed when exposed to air, the ratio of ODA to 6FDA was 1:1.02, and there was a slight excess of 6FDA. The excess of 6FDA was 2 mol% to ensure that the resulting polyimide reacted in accordance with the conditions we defined. After stirring, a polyamic acid solution (PAA) was made by placing the polymer solution and deforming it.

Solution casting was used to create polyimide membranes. In order to create a homogeneous membrane, the previously synthesized and acquired polyamic acid was evenly poured over a clean glass plate. In order to complete high-temperature cyclization and eliminate any remaining solvent after casting, the PAA was heated by gradient heating in a vacuum oven. This produced the polyimide membrane (PI). The precise temperatures used during the heating procedure were 80 °C for three hours, 120 °C for three hours, 160 °C for two hours, 200 °C for an hour, 240 °C for an hour, and 280 °C for an hour.

In Figure 1, it can be seen how the ionic liquid/polyimide hybrid membranes formed (PI/IL_n_ (x %)). The synthesis process of polyimide membranes with ionic liquid caps was illustrated using the PI/IL_1_ (5%) as an example. A three-necked flask was filled with DMF (5.60 mL), ODA (0.40 g, 2.00 mmol), and 6FDA (0.91 g, 2.04 mmol). The mixture was agitated for a further 4–5 h to ensure enough reaction before the addition of IL_1_ (0.02 g, 0.10 mmol) after an additional hour of stirring. After stirring, a polyamic acid solution (PAA/IL_1_ (5%)) was created by placing the polymer solution and deforming it. A similar technique to that used for PI was used to prepare the polyimide hybrid membranes via ionic liquid. 

### 2.3. Gas Separation Measurement and Calculation Formula

The gas permeation properties of mixed gas were determined on a variable-pressure constant-volume gas permeation cell. The dense membrane was mounted onto the permeation cell and vacuumed at 25 °C for 3 h before tests. The mixed gas permeation properties were evaluated using a binary mixture of 50/50 CO_2_/CH_4_ (0.33 nm), CH_4_ (0.38 nm). The membranes were tested at 25 °C with a feed pressure from 2 to 20 atm. The mixed gas composition was analyzed by a GTR-11MH series gas chromatograph (GC). The detailed experimental design and procedures have been reported by our group. The gas permeabilities of CO_2_ and CH_4_ are expressed by the following equations. The gas separation permeability of PI and PI/IL_n_ (x %) membranes was tested by gas chromatography (GTR-11MH, GTR TEC Corporation, Kyoto, Japan). The permeability (P), selectivity (α), diffusion coefficient (D), and dissolution coefficient (S) were calculated by the following equations. 

For the permeability (P), Equation (1) was applied:(1)$$P(Barrer)=\frac{q\times K\times L}{a\times p\times t}(mL\cdot cm\cdot {cm}^{-2}\cdot s^{-1}\cdot cmHg^{-1})$$
where q is the transport volume (mL), K represents the auxiliary positive coefficient (fixed value 2), L denotes the membrane thickness (cm), a is the area of the gas permeable membrane (fixed value 0.785 cm^2^), p designates the permeability pressure (cm Hg), and t is the measurement time (s).

The selectivity (α) was calculated using Equation (2):(2)$$\alpha =\frac{P{CO}_{2}}{P{CH}_{4}}$$

And the diffusion (D) and dissolution (S) coefficients were calculated using Equations (3) and (4), respectively:(3)$$D=\frac{L^{2}}{6T}$$
(4)$$S=\frac{P}{D}$$

In the above, L and T are the thickness and residence time of the membranes, respectively.

#### Other Measurements

For the purpose of ensuring the reproducibility of the experimental findings, all characterization data were collected in the membrane state using the average. The polyimide membrane’s surface morphology and surface structure were scanned using a scanning electron microscope (S3400N, Hitachi, Tokyo, Japan). To create the uniform polyimide membranes, a spin coater (SYSC-50, Shanghai Sanyan Technology Co., Ltd., Shanghai, China) was employed. The thickness of the polyimide membranes was measured using a membrane thickness gauge (CH-1-B, Shanghai Liuling Instrument Factory, Shanghai, China) with a graduation value of 0.001 mm (the measurement range was 0–1 mm, and the error was 0.001 mm), and the mechanical properties of the polyimide membranes were measured using a membrane tensile testing machine (XLW(PC)–500N, Sumspring, Jinan, China). 

## 3. Results and Discussion

### 3.1. Surface Morphology of PI and PI/IL_1_ (x %) Membranes

To observe the distribution of IL in a polyimide matrix, the membrane surface morphology was tested using a scanning electron microscope. The SEM photos of pure PI, PI/IL_1_ (5%), PI/IL_1_ (10%), PI/IL_1_ (15%), and PI/IL_1_ (20%) composites were given in Figure 1a–e. According to Figure 1, the surface of the pure polyimide membrane was dense and flat. The membrane surfaces of PI/IL_1_ (5%) and PI/IL_1_ (10%) (Figure 1b,c) were smoother and clearer than those of PI/IL_1_ (15%). IL was evenly dispersed in the polyimide matrix in PI/IL_1_ (15%) (Figure 1d), while the particle size was small and homogenous in PI/IL_1_ (20%) (Figure 1e). At the same time, in the blend membranes, it could be seen that there were obvious particle-like substances, and with the increase in blending content, the particle-like substances on the surface of the membranes significantly increased. As depicted in Figure 1, the distribution of IL in a polyimide matrix is homogeneous. Furthermore, as a result of the inherent high aspect ratio and blend by the high-speed mixer self-assembly process, these PI/IL_n_ align with each other in parallel to constitute a compactly uniform structure. Such a uniform structure creates many stable nanochannels for gas permeation. After spin-coating, the surface of PI/IL_n_ becomes smoother, as well as a denser cross-section, without any conspicuous voids.

### 3.2. Mechanical Properties of PI and PI/IL_n_ (x %) Membranes

A number of tensile tests were carried out to evaluate the mechanical attributes of the polyimide gas separation membranes. The test samples had the following specifications: a length of 30 mm, a width of 10 mm, and a test speed of 5.00 mm/min. Appendix A displays the information regarding these tests. 

(1)The effect of IL content on the mechanical properties of PI membranes

When the ODA-6FDA is used as a substrate of polymer, the content of the same ionic liquid has a significant impact on the mechanical properties of the membrane. The range of elongation at the break of ionic liquid blended polyimide membrane is 5.03–21.09%, which is higher than the elongation at the break of the pure membrane by 4.86%, and approximately increases by 1.03–4.34 times. The tensile strength range of ionic liquid blended polyimide membrane is 86–166 MPa, which is higher than pure film 44 MPa and approximately increases by 1.95–3.75 times. The reason for this may be that the introduction of IL causes the amide bond on PI to interact with the hydrogen atom on IL and to form hydrogen bonds (Figure 2). The presence of hydrogen bonds in the polymer increases the mechanical properties of the membrane. Therefore, the polyimide-blended membrane with ionic liquids has excellent mechanical properties.

As shown in Appendix A and Figure 3, the elongation at break and the tensile strength of the PI/IL_n_ (x wt%) membrane increase with the increase in IL_n_ when the IL_n_ doping amount is low by comparing it with pure polyimide films. The highest value is reached when the IL_n_ doping amount is 15%, and decreases when the IL_n_ doping amount is 20%. For example, when the film is ODA-6FDA/IL (15 wt%), the elastic modulus is 21.09% and the tensile strength is 151 MPa; when the membrane is ODA-6FDA/IL5 (20 wt%), the elastic modulus is 10.99% and the tensile strength is 95 MPa. Both are greater than the mechanical properties of pure film (ODA-6FDA), with an elastic modulus of 4.86% and a tensile strength of 44 MPa. The reason for this may be that when the load of IL_n_ is low, the interface interaction between the added IL_n_ and PI is good, thus improving the mechanical properties of the mixed matrix membrane to a certain extent by increasing hydrogen bonds in the polymer. But with the increase in the IL_n_ doping amount, the interfacial interaction force decreases, which easily forms stress concentration points within the membrane, which affect the mechanical properties of the mixed matrix membrane.

(2)The effect of IL types on the mechanical properties of PI membranes

When the polymer is on the same substrate (ODA-6FDA) and the content of ionic liquids is the same, the type of ionic liquid also has a significant impact on the mechanical properties of the membrane. In Appendix A and Figure 3, we can see that, in most cases, the elongation at break and tensile strength of the PI/IL_n_ (x wt%) membrane decrease first, then increase, and finally decrease with the increase in alkyl chains in IL cations. The maximum value is reached when the number of carbon chains of cations in an ionic liquid is eight. For example, the ionic liquid is 1-octyl-3-methylimidazolium tetrafluoroborate (IL_3_). For example, the elongation at the break of the membrane with ODA-6FDA/IL_3_ (15 wt%) is 21.09%, and the elongation at the break of the membrane with ODA-6FDA/IL_4_ (15 wt%) is 9.62%; the tensile strength of the ODA-6FDA/IL_3_ (15 wt%) membrane is 151 MPa, while the tensile strength of the ODA-6FDA/IL_4_ (15 wt%) membrane is 144 MPa. The reason for this may be that when the number of carbon chains of cations in the ionic liquid is eight, the compatibility between the ionic liquid and the polyimide matrix reaches its best, resulting in the best mechanical properties of the membrane. It may also be related to the binding ability of hydrogen bonds of the membrane, and the mechanical properties of the thin membrane prepared under the different interfacial interaction force greatly vary. Therefore, different ionic liquids have a significant impact on the mechanical properties of the polyimide membrane.

In order to investigate the effect of the polyimide structure on the mechanical properties, the study found that when the IL type was the same and the IL content was different, the elongation at the break and tensile strength of the membranes increased with the content increase in IL, and when the IL content was 15%, the elongation at break and the tensile strength of the membrane reached the highest; when the IL content was 20%, the elongation at the break and the tensile strength of the membranes decreased (Figure 3). Taking the polyimide membrane doped with IL_1_ as an example, the elongation at breakage of PI/IL_1_ (15%) (21.07%) was about four times as much as PI (4.86%), the tensile strength (131 MPa) of PI/IL_1_ (15%) was about three times as much as PI (44 MPa). The reason for this might be because the mechanical properties of the mixed matrix membrane had been somewhat improved when the loading amount of IL was modest, because there was extensive interfacial interaction between the additional IL and PI [41]. However, when the amount of IL doping increases, the interfacial interaction is forced to diminish and the stress concentration point is easily produced in the membrane, which has an impact on the mechanical characteristics of mixed matrix membranes [42]. When the content of IL was the same, the types of IL were different, and the elongation at the break and the tensile strength of the membranes were not much different, which may be due to the ionic liquid having good compatibility in the polyimide matrix.

### 3.3. Gas Separation Performance of PI and PI/IL_n_ (x %) Membranes

To compare the gas permeability of the PI and PI/IL_n_ (x %) membranes, experiments were conducted to test for CO_2_ and CH_4_. Appendix A summarizes the permeability (P) and selectivity (α) of the two sets of membranes. The study found that the CO_2_/CH_4_ selectivity of the membrane was increased with the increase in the content of IL (Figure 4b) with the same type of IL. When the content of IL reached 15%, the selectivity was the highest, and when the content of IL was 20%, the selectivity of the membrane for mixed gas decreased (Appendix A). Taking the polyimide membrane doped with IL_3_ as an example, the selectivity of PI/IL_3_ (15%) (α(*PCO*_2_*/PCH*_4_) = 180.55) was about 2.5 t ionic liquids from the imidazoleimes, as many as PI (α(*PCO*_2_*/PCH*_4_) = 73.54), which was related to the dissolution coefficient of the membrane for CO_2_. With the increase in IL content, the dissolution coefficient of the membrane to CO_2_ gradually increases. For example, the dissolution coefficient of the PI/IL_3_ (15%) membrane to CO_2_ (*SCO*_2_ = 0.88) was about six times that of the PI membrane (*SCO*_2_ = 0.14). The imidazole groups from the ionic liquid are responsible for the change in the solubility coefficients of CO_2_ and CH_4_, respectively. The interaction between CO_2_ and CH_4_ that can result in the formation of hydrogen bonds between the two gases is known as coupling. Ionic liquids from the imidazole group have the ability to break up this connection, enhance the rivalry between the two gases, and interact with CO_2_ to increase the gas’ solubility. In contrast, when the content of IL was higher than 15%, this behavior resulted from the plasticization of PI induced by IL, the clogging effect caused CO_2_ to decrease through diffusion and dissolution. The selectivity of the membranes for CO_2_ and CH_4_ improved with an increase in the substituent chain on the IL when the IL concentration and types were the same and different, respectively. When the IL contained eight carbons, the selectivity of the membrane for CO_2_ and CH_4_ was the highest. This might be because the solubility of the membrane for CO_2_ increased with the growth of the alkyl chains [43].

From Appendix A, we know that when the IL content is lower than 15%, the CO_2_ permeability first decreases (Figure 4a), which may be due to the fact that the small ion of IL in the gap of the polymer chain inhibited the flexibility of the polymer chain and led to a more compact structure [44]. And from the diffusion coefficient of the membrane, we can know that when the IL content is lower than 15%, the gas diffusion coefficient also rapidly decreases from the increase in IL. For instance, in Appendix A, the CO_2_ diffusion coefficient of the PI/IL_3_ (15%) membrane (*DCO*_2_ = 18.42) was roughly 73% lower than the CO_2_ diffusion coefficient of PI (*DCO*_2_ = 67.75). As a result, the decrease in gas permeability at 15 weight percent IL can be attributed to a decrease in gas diffusivity. Additionally, there were evenly distributed IL plasticizes PI. Plasticizers often have two properties, called diffusion and blocking effects. The clogging effect causes a decrease in diffusion and dissolution while the diffusion effect causes an increase in gas diffusivity. As a result, rather than the plasticizing effect, the decrease in gas diffusivity appears to be dependent on the blocking impact of the gas diffusion channel generated by the scattered IL in the PI matrix [40], hence the permeability of CO_2_ decreases. In contrast, when the content of IL was higher than 15%, the permeability of CO_2_ increases, which may be because too much IL forms in the domain, which more effectively promotes the permeability of CO_2_, increasing CO_2_ permeability [40,45,46]. And in the diffusion coefficient of the membrane, we can know that when the IL content is higher than 15%, the gas diffusion coefficient also rapidly increases from the increase in IL. For example, the diffusion coefficient of membrane PI/IL_1_ (20%) to CO_2_ (*DCO*_2_ = 19.10) is greater than that of PI/IL_1_ (15%) to CO_2_ (*DCO*_2_ = 18.45) in Appendix A. This behavior has resulted from the plasticization of PI induced by IL, so the permeability of CO_2_ increases. 

Robeson upper bound plots of the PI and PI/IL_n_ (x %) blend membranes for gas pairs of CO_2_/CH_4_ are presented in Figure 5. When IL was IL_3_ and the content was 15%, the separation effect of the membrane was the best and exceeded the curve of 2008. When the IL content exceeded 15%, the permeability of the membrane significantly increased compared with the pure membrane, which was close to the Robeson curve in 1991. This result demonstrated that the PI/IL blend membranes have good potential to improve the permeability and selectivity of CO_2_ and CH_4_.

## 4. Conclusions

As detailed, polyimide/ionic liquid blended membranes were successfully prepared. Ionic liquid was present in the mix membrane at several degrees: 5%, 10%, 15%, and 20%. Due to the plasticization of PI brought on by IL, which enhances CO_2_ permeability, this phenomenon can be observed. When the types of ionic liquids are the same, the elongation at break and tensile strength of PI/IL_n_ (x wt%) membranes increase with the increase in IL_n_. When the IL_n_ doping amount is low, they reach the highest value when the IL_n_ doping amount is 15 wt%. When the content of ionic liquid is the same, the number of carbon chains of cations in the ionic liquid is eight, which reaches the maximum value when the ionic liquid is 1-octyl-3-methylimidazolium tetrafluoroborate (IL_3_). The reason for this may be that when the number of carbon chains of cations in the ionic liquid is eight, the compatibility between the ionic liquid and the polyimide matrix reaches its maximum, resulting in the best mechanical properties of the membrane. Additionally, there were many forms of IL; as the substituent chain length on the IL increased, so did the membranes’ selectivity for CO_2_ and CH_4_. According to the findings, the maximum PI/IL_3_ membrane separation coefficient (15%) was 180.55, above the 2008 curve.

## Data Availability

Data are contained within the article.

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
