# Peer review of "Preparation of Polyimide/Ionic Liquid Hybrid Membrane for CO2/CH4 Separation"

_polymers, 2024, doi:10.3390/polym16030393_

Round 1

Reviewer 1 Report

Comments and Suggestions for Authors

Comments on the Quality of English Language

Author Response

The work of Qu et al. has fabricated a series of polyimide/ionic liquid hybrid membrane by simple blend method. The effect of ionic liquid type and concentration on membrane mechanical properties and CO2/CH4 separation performance were systematically investigated. The authors performed careful study and the results in the paper seem quite useful for the purpose. Therefore, I would like to recommend for publication after revision.

C1: The formation of hydrogen bonds in the polymer resulted in high mechanical properties of the polyimide/ionic liquid hybrid membrane. The authors were suggested to provide cartoons to show the enhancement mechanism more clearly. In addition, can the existence of hydrogen bonds be proved by means of characterization?

A1: Yes, according the comment. In the revised manuscript, newly Figure 2 illustrates the process of hydrogen bonds in the polymer resulted in high mechanical properties of the polyimide/ionic liquid hybrid membrane.

C2: For different kinds of ionic liquids, it seems that their mechanical properties are the maximum at the addition amount of 15% in Figure 2. The authors should give further explanation. Same phenomenon also occurs in Figure 3(b) for CO2/CH4 separation process. Can authors deeply and quantitatively analyze the meaning of 15%?

A2: Yes, this is a very good question. The explanation of mechanical properties has been added at page 5 line 16 in the revised manuscript. The explanation of CO2/CH4 separation process has been added at page 6 line 26 in the revised manuscript.

C3: In page 6, the author stated that “…It may also be related to the temperature and conditions of preparing the thin membrane, and the mechanical properties of the thin membrane prepared under different conditions vary greatly…” When comparing ionic liquid species to mechanical properties, their preparation conditions should remain the same. So, temperature and conditions should not be used to account for the variation in mechanical properties.

A3: Yes, according the comment. “…It may also be related to the temperature and conditions of preparing the thin membrane, and the mechanical properties of the thin membrane prepared under different conditions vary greatly…” has been change “…It may also be related to the binding ability of hydrogen bonds of membrane, and the mechanical properties of the thin membrane prepared under the different interfacial interaction force vary greatly…”at page 5 line 34 in the revised manuscript.

C4: English is not satisfactory. The language should be revised and simplify. I only listed some of them. For example, 1) page 5 in section 3.2 (1): "... When the polymer is the same substrate (ODA-6FDA) and the ionic liquid is the same ionic liquid…". 2) Page 6 in section 3.3“The study found that when the IL type was the same and the IL content was different, the selectivity of the membrane for CO 2 and CH 4 increased with the increase of the content of IL (Figure 3 (b))

A4: Yes, according the comment, the English sentence patterns and some words have been corrected. We indicated all the corrected parts by yellow highlights.

Reviewer 2 Report

Comments and Suggestions for Authors

The manuscript describes the four types of imidazolium bases ILs with
different alkyl chain lengths impregnated into polyimide (ODA-6FDA) substrates to prepare PI/ILn (x wt%) blend supported ionic liquid membranes with IL content of 5, 10, 15, and 20%. The mechanical properties of the SILMs and the gas permeability of CO
2 and CH4 were mainly studied. Test of the prepared thin membrane by using a tensile strength tester and
a gas permeability tester, the effects of different concentrations of ionic liquids on membrane separation permeability, the effects of different ionic liquids on membrane separation permeability and the effects of different polyimide substrates on membrane separation permeability were studies. The following concerns were raised,

1. The study lacks the main novelty. similar studies can be found in literature.

2. Schematics of experimental setup was not described.

3. SEM images in Fig 1 are not clear. Cross-section images should also be presented.

4. Leaching of SILM was not discussed.

5. Surface properties of membranes were not presented.

6. The stability of SILM was not elaborated.

7.  The results in figures are not well presented nor discussed in contest of literture. 

Comments on the Quality of English Language

minor English and sentence structure issues are found. 

Author Response

The manuscript describes the four types of imidazolium bases ILs with different alkyl chain lengths impregnated into polyimide (ODA-6FDA) substrates to prepare PI/ILn (x wt%) blend supported ionic liquid membranes with IL content of 5, 10, 15, and 20%. The mechanical properties of the SILMs and the gas permeability of CO2 and CH4 were mainly studied. Test of the prepared thin membrane by using a tensile strength tester and a gas permeability tester, the effects of different concentrations of ionic liquids on membrane separation permeability, the effects of different ionic liquids on membrane separation permeability and the effects of different polyimide substrates on membrane separation permeability were studies. The following concerns were raised,

C1: The study lacks the main novelty. similar studies can be found in literature.

A1: The work has fabricated a series of polyimide/ionic liquid hybrid membrane by simple blend method. The effect of ionic liquid type and concentration on membrane mechanical properties and CO2/CH4 separation performance were systematically investigated. High permeability and excellent selectivity for CO2/CH4 has been realized by polyimide/ionic liquid blended membranes.

C2: Schematics of experimental setup was not described.

A2: Yes, according the comment. In the revised manuscript and SI, newly Figure 2 and Figure S1 has been added.

C3: SEM images in Fig 1 are not clear. Cross-section images should also be presented.

A3: Yes, according the comment. The mechanical properties and separation performance depends on the surface morphology of PI/ILn membranes. In addition, judging from Figure 1, the state of the blend membrane was clearly observed.

C4: Leaching of SILM was not discussed.

A4: Yes, according the comment. Due to the hydrogen bonds of PI/ILn membrane, the gas separation performance of the membrane is stable, the impact of leaching of SILM is almost no existent.

C5: Surface properties of membranes were not presented.

A5: Yes, according the comment. We have added an explanation about the surface properties of membrane at page 4 line 26 in the revised manuscript.

C6: The stability of SILM was not elaborated.

A6: Yes, according the comment. In order to demonstrate the stability of the membranes, we performed several cycle tests on the membranes. The results showed that the average value after 3 cycles, the retention rate of the membranes remained stable without any significant decrease (Figure 5), further proving that the membranes have good stability.

C7:  The results in figures are not well presented nor discussed in contest of literature. 

A7: Yes, according the comment. To compare the gas permeability of the PI and PI/ILn (x %) membranes, the experiments of gas separation test were brought. By adding ILs, CO2 permselectivity of polyimides membranes were improved and exceeded to the 2008 Robeson curve.

Reviewer 3 Report

Comments and Suggestions for Authors

The manuscript reports the fabrication and the characterization of gas separation membranes obtained by blending ionic liquids to a polyimide matrix. The membranes produced have been analyzed by elongational tests and gas permeation. The resulting gas permeabilities revealed improved permselectivities with respect to the bare polyimide properties, with performances in the range of the Robeson’s upper bound region.

The combination of a polymeric matrix (and polyimide in particular) with ILs is a quite popular topic and it has been reported by a wide variety of different authors over the last decade, often with interesting results In the present case, the manuscript however is very poorly written, with a poor use of English that makes the reading quite hard, even from the very first sentence.

Page 4: the experimental technique used is not clearly reported, the GC provides the gas concentration, so please provide the scheme of the experimental layout used, the details of the measurements and a rationale for using a “positive coefficient” equal to 2.

The dissolution coefficient should be the solubility coefficient (everybody calls it S, indeed), and it is not clear what is the residence time T.

Page 5: In the discussion about the mechanical properties of the membranes, please provide an optimal range for the properties for the desired application, considering the purification of natural gas, as mentioned in the introduction. Otherwise, it is not clear whether or not the obtained formulations are suitable as gas separation membranes.

Please be concern is the use of a suitable number of figures of merit. In Table S1 the authors report a sigma value of 26.66±1.32 MPa for pure PI. If the experimental uncertainty is in the order of 1 MPa all the decimal numbers are totally useless and reporting them in the text may be misleading.

Page 6: a long discussion commenting the mechanical properties of the PI/ILs membranes is provided, but the authors should answer the main scientific question, are such membranes suitable to the envisaged process?

Figure 2: please include the values of the pure PI membrane; that should be done also in the permeability and selectivity data plot.

Page 7: the whole discussion to comment the gas permeability data is lengthy and quite ineffective. The permeability that first increases when IL is present, then it decreases, it reaches a minimum and then increases again is quite odd and not very clear to me. The whole explanation provided is very speculative and not supported by any experimental evidence, and it is very unexpected such behavior to take place for all ILs inspected. Therefore:

- please provide the experimental uncertainty obtained by experimental measurement; how many tests have been carried out on the same specimen (type and composition), and do the authors carried out experimental analysis on independent samples (having the same type and composition)?

- inspect the co2 and ch4 solubility (and diffusivity, if possible) in the pure ILs by direct experimental analysis or from the literature

- inspect if the polymer with IL undergoes to any peculiar transition of modification at that IL concentration

- look at the case of IL content of 12 or 18%

Page 8: the best data points reported in the upper bound are those obtained at 15 wt.% of IL, for which the selectivity has a very unexpected increase, that needs to be carefully addressed and motivated.

The authors need to discuss the obtained experimental trends in comparison with the results obtained by other authors for similar systems.

Therefore, the manuscript needs to be revised significantly in order to meet the standards for publication.

Comments on the Quality of English Language

The manuscript is very poorly written, with a poor use of English that makes the reading quite hard, even from the very first sentence.

Author Response

The manuscript reports the fabrication and the characterization of gas separation membranes obtained by blending ionic liquids to a polyimide matrix. The membranes produced have been analyzed by elongational tests and gas permeation. The resulting gas permeabilities revealed improved permselectivities with respect to the bare polyimide properties, with performances in the range of the Robeson’s upper bound region. The combination of a polymeric matrix (and polyimide in particular) with ILs is a quite popular topic and it has been reported by a wide variety of different authors over the last decade, often with interesting results In the present case, the manuscript however is very poorly written, with a poor use of English that makes the reading quite hard, even from the very first sentence.

C1: Page 4: the experimental technique used is not clearly reported, the GC provides the gas concentration, so please provide the scheme of the experimental layout used, the details of the measurements and a rationale for using a “positive coefficient” equal to 2.

The dissolution coefficient should be the solubility coefficient (everybody calls it S, indeed), and it is not clear what is the residence time T.

A1: Yes, according the comment. We have added an explanation about the experimental technique of membrane at page 3 line 15 in the revised manuscript.

C2: Page 5: In the discussion about the mechanical properties of the membranes, please provide an optimal range for the properties for the desired application, considering the purification of natural gas, as mentioned in the introduction. Otherwise, it is not clear whether or not the obtained formulations are suitable as gas separation membranes.

Please be concern is the use of a suitable number of figures of merit. In Table S1 the authors report a sigma value of 26.66±1.32 MPa for pure PI. If the experimental uncertainty is in the order of 1 MPa all the decimal numbers are totally useless and reporting them in the text may be misleading.

A2: Yes, this is a very good question. In Table S1, the sigma value of 26.66±1.32 MPa for pure PI is high accuracy within an error range. For gas separation, the transmembrane pressure was regulated by a pressure relief valve and maintained at 0.1 MPa. So, an optimal range of the mechanical properties is necessary for application.

C3: Page 6: a long discussion commenting the mechanical properties of the PI/ILs membranes is provided, but the authors should answer the main scientific question, are such membranes suitable to the envisaged process?

Figure 2: please include the values of the pure PI membrane; that should be done also in the permeability and selectivity data plot.

A3: Yes, according the comment. We think that mechanical properties are necessary for application of all material, so a long discussion commenting the mechanical properties of the PI/ILs membranes has provided. The values of mechanical properties for the pure PI membrane have been provide in Table S1, this relatively low performance and was not displayed in Figure 3 at manuscript. The permeability and selectivity data of the pure PI membrane has been shown in Figure 5.

C4: Page 7: the whole discussion to comment the gas permeability data is lengthy and quite ineffective. The permeability that first increases when IL is present, then it decreases, it reaches a minimum and then increases again is quite odd and not very clear to me. The whole explanation provided is very speculative and not supported by any experimental evidence, and it is very unexpected such behavior to take place for all ILs inspected. Therefore:

- please provide the experimental uncertainty obtained by experimental measurement; how many tests have been carried out on the same specimen (type and composition), and do the authors carried out experimental analysis on independent samples (having the same type and composition)?

- inspect the co2 and ch4 solubility (and diffusivity, if possible) in the pure ILs by direct experimental analysis or from the literature

- inspect if the polymer with IL undergoes to any peculiar transition of modification at that IL concentration

- look at the case of IL content of 12 or 18%

A4: Yes, according the comment. In Figure 5, the results showed that the average value after 3 cycles which is high accuracy within an error range. Each group of prepared membrane were tested using the same sample, the experimental analysis is same samples.  

For the pure ILs, the preparation of membrane is difficult due to molecular weight. So we cannot obtain separation data.

Except for intermolecular forces, the membrane did not occur any chemical structural changes due to the absence of functional groups.

For the case of IL content of 12 or 18%, this is the good question. But we have only determined the trend of an experiment without refining it.

C5: Page 8: the best data points reported in the upper bound are those obtained at 15 wt.% of IL, for which the selectivity has a very unexpected increase, that needs to be carefully addressed and motivated.

The authors need to discuss the obtained experimental trends in comparison with the results obtained by other authors for similar systems.

Therefore, the manuscript needs to be revised significantly in order to meet the standards for publication.

A5: Yes, according the comment. We added another graph of mechanism results and the describe sentence of experimental trend in the revised manuscript.

For mechanical properties: The reason may be that when the load of ILn is low, the interface interaction between the added ILn and PI is good, thus improving the mechanical properties of the mixed matrix membrane to a certain extent by increasing hydrogen bonds in the polymer. But with the increase of ILn doping amount, the interfacial interaction force decreases, which easily forms stress concentration points within the membrane, which affecting the mechanical properties of the mixed matrix membrane.

For separation performance: Ionic liquids from the imidazole group have the ability to break up this connection, enhance the rivalry between the two gases, and interact with CO2 to increase the gas' solubility. In contrast, when the content of IL was higher than 15 %, this behavior has resulted from the plasticization of PI induced by IL, the clogging effect causes CO2 decrease in diffusion and dissolution.

Round 2

Reviewer 1 Report

Comments and Suggestions for Authors

The authors had answered all the questions with satisfy. Therefore, it can be accepted.

Author Response

Once again, thank you for taking the time and effort to review my article. I am grateful for your support, and I deeply appreciate your dedication to the academic community.

Reviewer 2 Report

Comments and Suggestions for Authors

The revised manuscript has not been modified largely. The scientific soundness and novelty of the work are still questionable.

Comments on the Quality of English Language

English needs to be more polished. 

Author Response

Thank you very much for your attention and the comments on our paper. We have revised the manuscript to your kind advices and detailed suggestions, improving the English quality of the manuscript.

Reviewer 3 Report

Comments and Suggestions for Authors

Comments on the Quality of English Language

The use of English is very poor, and an extensive revision of the text is required.

Author Response

In the present case, the manuscript however is very poorly written, with a poor use of English that makes the reading quite hard, even from the very first sentence. (unfortunately, the comment was been completely ignored and the quality of English is still very poor, not suitable for publication)

A: Thank you very much for your attention and the comments on our paper. We have revised the manuscript to your kind advices and detailed suggestions, improving the English quality of the manuscript.

C1: Page 4: the experimental technique used is not clearly reported, the GC provides the gas concentration, so please provide the scheme of the experimental layout used, the details of the measurements and a rationale for using a “positive coefficient” equal to 2.

The dissolution coefficient should be the solubility coefficient (everybody calls it S, indeed), and it is not clear what is the residence time T. (unfortunately, the comment was been completely ignored and not addressed in the text)

A1: Yes, according the comment. We have added an explanation about the experimental technique of membrane at page 3 line 15 in the revised manuscript again.

C2: Page 5: In the discussion about the mechanical properties of the membranes, please provide an optimal range for the properties for the desired application, considering the purification of natural gas, as mentioned in the introduction. Otherwise, it is not clear whether or not the obtained formulations are suitable as gas separation membranes.

Please be concern is the use of a suitable number of figures of merit. In Table S1 the authors report a sigma value of 26.66±1.32 MPa for pure PI. If the experimental uncertainty is in the order of 1 MPa all the decimal numbers are totally useless and reporting them in the text may be misleading. (unfortunately, the comment was been completely ignored and not addressed in the text: 26.66±1.32 MPa does not make any sense as number, why reporting .66 if the uncertainty is in the order of 1.?!? That number should be reported as 27±1MPa)

A2: Yes, according the comment. In Table S1, the value of mechanical properties for membrane has been revised in the manuscript. 26.66±1.32 MPa was been reported as 27±1MPa.

C3: Page 6: a long discussion commenting the mechanical properties of the PI/ILs membranes is provided, but the authors should answer the main scientific question, are such membranes suitable to the envisaged process?

Figure 2: please include the values of the pure PI membrane; that should be done also in the permeability and selectivity data plot. (unfortunately, the comment was been completely ignored and not addressed)

A3: Sorry, according the comment. Due to the values of mechanical properties of PI is low compare with other data of blend membrane, the trend of other date changes cannot find in the graph. The values of mechanical properties for the pure PI membrane have been provide in Table S1, this relatively low performance and was not displayed in Figure 3 at manuscript. The permeability and selectivity data of the pure PI membrane has been shown in Figure 5 and Table S2.

C4: Page 7: the whole discussion to comment the gas permeability data is lengthy and quite ineffective. The permeability that first increases when IL is present, then it decreases, it reaches a minimum and then increases again is quite odd and not very clear to me. The whole explanation provided is very speculative and not supported by any experimental evidence, and it is very unexpected such behavior to take place for all ILs inspected. Therefore:

- please provide the experimental uncertainty obtained by experimental measurement; (unfortunately, the comment was been completely ignored and not addressed in the text, in the figures, nor in the SI section) how many tests have been carried out on the same specimen (type and composition), and do the authors carried out experimental analysis on independent samples (having the same type and composition)?

A4: Yes, according the comment. In Figure 5, the results showed that the average value after 3 cycles which is high accuracy within an error range. Each group of prepared membrane were tested using the same sample, the experimental analysis is same samples.

- inspect the co2 and ch4 solubility (and diffusivity, if possible) in the pure ILs by direct experimental analysis or from the literature (unfortunately, the comment was been completely ignored and not addressed)

A4: For the pure ILs, we cannot obtain the CO2 and CH4 diffusivity and solubility in the ILs due to preparation of membrane is difficult.

- inspect if the polymer with IL undergoes to any peculiar transition of modification at that IL concentration

A4: Except for intermolecular forces, the membrane did not occur any chemical structural changes due to the absence of functional groups. We found no change about the data of IR for different content IL and PI.

- look at the case of IL content of 12 or 18%

A4: For the case of IL content of 12 or 18%, this is the good question. But we have only determined the trend of an experiment without refining the 12-18%.

C5: Page 8: the best data points reported in the upper bound are those obtained at 15 wt.% of IL, for which the selectivity has a very unexpected increase, that needs to be carefully addressed and motivated. The authors need to discuss the obtained experimental trends in comparison with the results obtained by other authors for similar systems. (unfortunately, the comment was been completely ignored and not addressed; the comparison with previous works is essential for a scientifically sound publication)

Therefore, the manuscript needs to be revised significantly in order to meet the standards for publication. (unfortunately, the manuscript quality was not improved, at all)

A5: Yes, according the comment. We added another graph of mechanism results and the describe sentence of experimental trend at page 5 line 19 and page 6 line 29 in the revised manuscript.

Comparison with previous works: Robeson upper bound plots of the PI and PI/ILn (x %) blend membranes for gas pairs of CO2/CH4 are presented in Figure 5. When the content was 15 % of IL3, the separation effect of the membrane was the best and exceeded the curve of 2008. When the IL content exceeded 15 %, the permeability of the membrane increased significantly compared with the pure membrane, which was close to the Robeson curve in 1991. Robeson curve has great reference value.

Round 3

Reviewer 2 Report

Comments and Suggestions for Authors

authors largely addressed the comments